# Optimization and Experimental Study of Structural Parameters for a Low-Damage Packing Device on an Apple Harvesting Platform

**Zixu Chen** [1], **Hongjian Zhang** [1,2], **Huawei Yang** [1,3], **Yinfa Yan** [1,4], **Jingwei Sun** [1], **Guangze Zhao** [1], **Jinxing Wang** [1,2,*] and **Guoqiang Fan** [1,4,*]

[1] College of Mechanical and Electrical Engineering, Shandong Agricultural University, Taian 271002, China; 2021120638@sdau.edu.cn (Z.C.); zhanghongji_an@163.com (H.Z.); 2021110437@sdau.edu.cn (H.Y.); guanghaok@gmail.com (Y.Y.); sunjw0714@163.com (J.S.); 17861501538@163.com (G.Z.)
[2] Shandong Provincial Key Laboratory of Horticultural Machinery and Equipment, Taian 271018, China
[3] Shandong Academy of Agricultural Machinery Sciences, Jinan 250010, China
[4] Shandong Agricultural Equipment Intelligent Engineering Laboratory, Taian 271002, China
* Correspondence: jinxingw@163.com (J.W.); fgq1217@163.com (G.F.)

**Abstract:** To address the issues of low efficiency and high damage rates during apple harvesting and packing, a parameter optimization experiment was conducted on a low-damage packing device for an apple harvesting platform based on Adams 2019 software. The aim was to reduce the mechanical damage to apples during the packing process. Firstly, kinematics and energetics analyses of the apple packing process were performed, and a mathematical model for damage energy was established to identify the main factors and their ranges that influence the mechanical damage to apples. Secondly, using the fruit damage rate and packing efficiency as the evaluation criteria, a second-order orthogonal rotating regression experiment was conducted with the inclination angle of the fruit conveying tube, the inner wall radius of the fruit conveying tube, and the length of the fruit conveying tube as the experimental factors. Regression mathematical models were established to assess the relationship between the evaluation criteria and the experimental factors. Finally, the impact of each experimental factor on the evaluation criteria was analyzed to determine the optimal structural parameters for the low-damage packing device of the apple harvesting platform, and validation experiments were conducted. The results showed that when the inclination angle of the fruit conveying tube was 47°, the inner wall radius of the fruit conveying tube was 84 mm and the length of the fruit conveying tube was 0.12 m, the average fruit damage rate was minimized at 7.2%, and the average packing efficiency was maximized at 1925 kg/h. These results meet the requirements for apple harvesting operations, and the research findings can serve as a reference for the structural design and packing operation parameter optimization of apple harvesting platforms.

**Keywords:** apple; harvesting platform; low-damage packing device; mechanical damage; parameter optimization

## 1. Introduction

China is the world's largest country in terms of their apple cultivation area and production rates. As of 2021, China's apple cultivation area reached 31 million hectares, with a production quantity of 45.97 million tons [1]. Apples provide essential nutrients such as vitamins, dietary fiber, and minerals to the human body, making them highly nutritious. Considering the retention of nutrients, the fresh consumption and low-level processing of apples are becoming increasingly popular. Consequently, the demand for fresh consumption and minimally processed apples has surged. To meet this demand, large-scale apple cultivation and mechanized management techniques are necessary. However, apple flesh is highly susceptible to damage, especially during the mechanized harvesting

process. Mechanical damage during harvesting can lead to structural disruptions in apple flesh cells, increasing the susceptibility to microbial decay and adversely affecting the storage of fresh apples [2]. Therefore, reducing the mechanical damage during apple harvesting is of significant importance in preserving the nutritional quality of apples.

Scholars from both domestic and international backgrounds have conducted diverse experimental studies on fruit damage. Wang et al. focused on pears and established a correlation between pear damage and vibration levels, revealing that the mechanical damage to pears during conveying is related to their placement method and stacking position. Van Linden et al. conducted impact experiments with varying intensities and found that the bruising potential of low- and medium-energy impacts primarily depends on the fruit texture, while the high-energy impacts' bruising potential depends on the fruit ripeness and impact location. Van Zeebroeck et al. developed a bruise prediction model through impact experiments, demonstrating that factors such as the impact energy, cultivation variety, ripeness, impact location, and curvature radius significantly influence the fruit's susceptibility to bruising. Yazdani et al. conducted free-fall experiments and established a multiple linear regression model between fruit the bruise volume and impact acceleration energy and deformation on different cushioning surfaces. Ishikawa et al. performed apple drop experiments, indicating that the percentage of damaged apples in single-wall and double-wall corrugated boxes increases with the drop height during conveying, with lower layer apples experiencing more significant damage than upper layer apples [3–6]. En et al. introduced an observer-based robust tracking control method, which effectively enables collision-free cooperative formation tracking control within a group of combine harvesters. Stopa et al. have developed a method that determines bruise resistance using average surface pressures as a load parameter, correlated to the volume of damaged tissue. Komarnicki et al. proposed a graphical approach for assessing bruise resistance and the bruise threshold. This method serves as an effective tool for evaluating mechanical damage [7–10]. Lixin et al. proposed a nonlinear dynamic rheological model and investigated the impact of the vibration intensity, cushioning packing structure, and layer placement on the apple damage. Zhen et al. designed a traction-type fruit orchard picking platform conveying system and simulation experiments that indicated that the connection method and cushioning materials of the fruit conveying system are the main limiting factors in reducing the fruit damage rate. Yudong et al. focused on blueberries and established a vibration model for blueberries, conducting a mechanism analysis of mechanical damage during blueberry conveyance, with the vibration of the conveying system itself being identified as the main cause of blueberry damage [11–13].

In summary, fruits are susceptible to damage during mechanical harvesting. Scholars from both domestic and international backgrounds have mainly focused on the mechanical properties of the fruits themselves and the structural aspects of cushioning materials in studying the mechanical damage during fruit harvesting. However, research on the structural design and parameter configuration of harvesting platforms is relatively scarce. In view of these reasons, this study analyzed the mechanism of mechanical damage during the packing process of apples and conducted parameter optimization and experiments on low-damage packing devices for an apple harvesting platform. The research identified the optimal structural design and operational parameters for low-damage packing devices, providing valuable insights for the structural design and optimization of operational parameters in apple harvesting platforms.

## 2. Materials and Methods

### 2.1. Overall Structure and Work Principle

The apple harvesting platform mainly consists of a driving chassis, picking workstations, terminal conveying devices, an aggregate conveying device, low-damage packing devices, and fruit boxes. The low-damage packing device includes a lifting frame, a belt, a fruit conveying tube, fruit trays, and brush rollers, as shown in Figure 1.

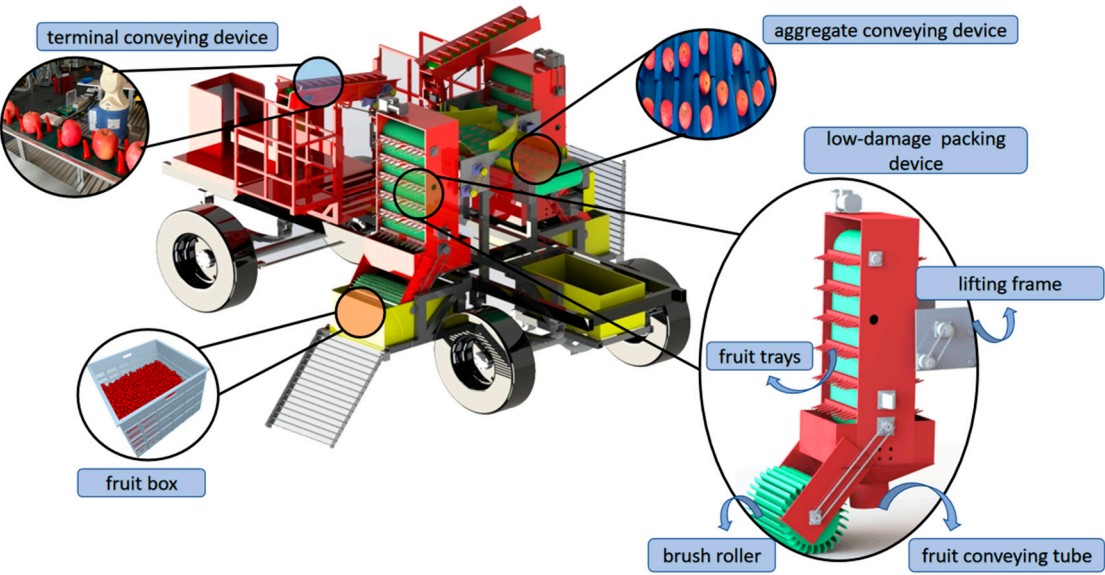

**Figure 1.** The overall structure of the apple harvesting platform.

Before operating the apple harvesting platform, workers adjust the position of the picking workstations and the opening and lifting angles of the terminal conveying device based on the orchard planting spacing and the growth status of the fruit trees. The preset range of belt conveyor speeds is also determined. During the packing operation, the harvested apples are placed on the terminal conveying device and conveyed to the aggregate conveying device. The apples of different sizes and grades pass through screening baffles of different heights and enter the low-damage packing devices on both sides. Subsequently, the apples fall into the fruit boxes through the fruit conveying tubes, while the brush rollers ensure the even distribution of the apples inside the boxes, enabling integrated operations of conveyance, size grading, and low-damage packing.

### 2.2. Main Technical Parameters

The low-damage packing device is primarily used for the packing operation on the apple harvesting platform, and its main parameters are listed in Table 1 below.

**Table 1.** Main technical parameters of the low-damage packing device.

| Technical Parameters | Level |
|---|---|
| Length × width × height/(mm × mm × mm) | 1040 × 480 × 410 |
| Matched power/kW | ≥17 |
| Belt linear speed/(m/s) | 0–0.6 |
| Adjustable range of the inclination angle of the fruit conveying tube/° | 0–90 |
| Adjustable range of the length of the fruit conveying tube/(m) | 0.12–0.30 |
| Adjustable range of the inner wall radius of the fruit conveying tube/(mm) | 60–120 |
| Material density of the fruit conveying tube/(kg/m³) | 965 |

### 2.3. Analysis of the Apple Packing Collision Process

When apples rotate with the fruit tray, they slide along the fruit tray under the influence of gravity, collide with the upper end of the fruit conveying tube, and then slide down along the inner wall of the fruit conveying tube before being propelled into the fruit box at the end of the tube. Based on the analysis of the collision process described above, the main stages of apple damage during the packing process include friction between the apple and the fruit tray during sliding off the tray, collision between the apple and the upper end of the fruit conveying tube, and friction between the apple and the inner wall of the fruit conveying tube when sliding down. Since the length of the fruit tray is relatively

short, the damage caused by friction between the apple and the fruit tray is relatively small compared to the damage caused in the latter two stages. Therefore, this paper does not analyze the friction between the apple and the fruit tray. The process of apple damage can be summarized as the collision between the apple and the upper end of the fruit conveying tube, the friction between the apple and the inner wall of the fruit conveying tube during sliding, and apples experiencing collisions upon falling into the fruit box. The specific process is shown in Figure 2.

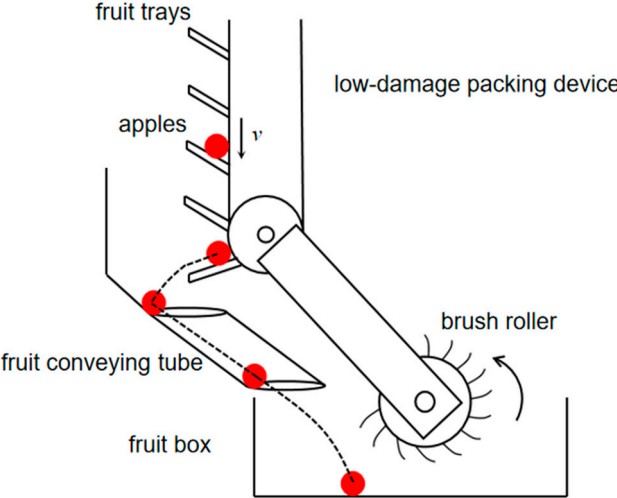

**Figure 2.** The apple movement process.

*2.4. Kinematic Analysis of the Packing Collision Process*

When the fruit tray undergoes a negative angle with respect to the ground, the apple starts to slide downward along the fruit tray under the influence of gravity with a constant acceleration $a_1$. The velocity of the apple is in the same direction as the fruit tray. Once the apple detaches from the fruit tray, it follows a parabolic trajectory, with a constant horizontal velocity and a vertical acceleration of $g$, until it collides with the inner wall of the fruit conveying tube. The kinematic analysis of the collision process between the apple and the fruit conveying tube is depicted in Figure 3.

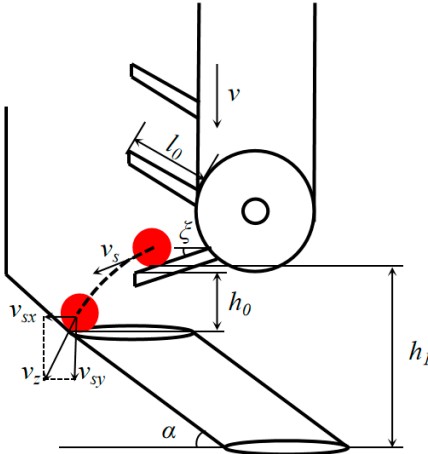

**Figure 3.** Kinematic analysis of the collision process between the apple and the fruit conveying tube.

The line speed $v$ range for the low-damage packing device is 0–0.6 m/s. Based on the actual operational conditions in the orchard, this paper adopts a velocity of 0.2 m/s. The length of the fruit tray $l_0$ is 0.12 m. When the fruit tray forms a negative angle $\xi$ with the horizontal plane, the apple starts to slide along the fruit conveying tube. Due to the relatively small angular velocity of the fruit tray, the sliding duration of the apple along the

fruit tray is short. Throughout the process, the variation of the angle between the fruit tray and the horizontal plane is negligible. Therefore, the change in the angle between the fruit tray and the horizontal plane during the sliding process of the apple along the fruit tray is ignored.

The instantaneous velocity of the apple upon detachment from the fruit tray is in the same direction as the fruit tray, and its magnitude is calculated as follows:

$$v_s = \sqrt{2l_0 g \sin\xi - (v\sin\gamma)^2} \tag{1}$$

where $v_s$ is the instantaneous velocity of the apple when it separates from the fruit tray, m/s; $l_0$ is the length of the fruit tray, m; $g$ is acceleration due to gravity; m/s$^2$; $\xi$ is the fixed angle between the fruit tray and the horizontal plane, assumed to be $-30°$; $v$ is the line speed range for the low-damage packing device; m/s; $\gamma$ is the inclination angle of the fruit tray, °.

The vertical component of the velocity when the apple collides with the fruit conveying tube is calculated as follows:

$$v_{sy} = \sqrt{2h_0 g + (v_s \sin\xi)^2} \tag{2}$$

where $v_{sy}$ is the instantaneous vertical velocity of the collision between the apple and the fruit conveying tube, m/s; $h_0$ is the vertical height of the apple in projectile motion, m.

The horizontal component of the velocity when the apple collides with the fruit conveying tube is calculated as follows:

$$v_{sx} = v_s \cos\xi \tag{3}$$

where $v_{sx}$ is the instantaneous horizontal velocity of the collision between the apple and the fruit conveying tube, m/s.

By substituting the numerical values, it can be calculated that $v_{sy} = 1.38$ m/s and $v_{sx} = 0.94$ m/s.

The final collision velocity $v_z$ between the apple and the inner wall of the fruit conveying tube is calculated as follows:

$$v_z = \sqrt{v_{sx}^2 + v_{sy}^2} \tag{4}$$

By substituting the numerical values, it can be calculated that $v_z = 1.67$ m/s.

During the downward sliding process of the apple along the fruit conveying tube, the longitudinal cross-section of the fruit conveying tube is considered as the sliding plane for the apple, and the apple itself is treated as a particle. The rolling motion of the apple and the influence of air resistance are neglected. The entire motion of the apple, from the start of sliding to the moment it detaches from the fruit conveying tube, is considered to occur within a single plane. The schematic diagram of the apple's downward sliding trajectory along the tube is shown in Figure 4.

Using the fruit conveying tube as a reference, with the center of gravity of the apple at the moment of collision with the inner wall of the tube as the coordinate system origin O, a Cartesian coordinate system is established with the positive *X*-axis pointing to the right horizontally and the positive *Y*-axis pointing upward vertically. The instantaneous horizontal velocity and vertical velocity of the apple as it slides along the inner wall of the tube until it detaches from the tube are $v_{tx}$ and $v_{ty}$:

$$\begin{cases} v_{tx} = \cos\alpha \sqrt{2g(\sin\alpha - \mu\cos\alpha)l_1} \\ v_{ty} = \sin\alpha \sqrt{2g(\sin\alpha - \mu\cos\alpha)l_1} \end{cases} \tag{5}$$

where $v_{tx}$ is the instant horizontal velocity of the apple upon detachment from the fruit conveying tube, m/s; $v_{ty}$ is the instant vertical velocity of the apple upon detachment from

the fruit conveying tube, m/s; $\mu$ is the coefficient of friction; $l_1$ is the length of the fruit conveying tube, m; $\alpha$ is the inclination angle of the fruit conveyor tube, $°$.

The expressions for the horizontal and vertical displacements are:

$$\begin{cases} f_x = \frac{1}{4}g\left(sin2\alpha - 2\mu cos^2\alpha\right)t^2 \\ f_y = \frac{1}{2}g\left(sin^2\alpha - \frac{1}{2}\mu sin2\alpha\right)t^2 \end{cases} \tag{6}$$

where $t$ is the contact time between the apple and the inner wall of the fruit conveying tube, s.

According to the above equation, it can be observed that the instant velocity of the apple upon detachment from the fruit conveying tube is primarily influenced by the inclination angle and length of the tube when the friction coefficient of the tube's inner wall remains constant. To ensure smooth packing of the apples without blockages during their passage through the tube, the inner wall radius of the tube should be proportional to the transverse diameter of the apples:

$$R_2 = \lambda R_1 \tag{7}$$

where $R_2$ is the inner wall radius of the fruit conveying tube, mm; $\lambda$ is the proportionality coefficient, taken within the range of 1.8 to 2.5; $R_1$ is the apple's transverse diameter, mm.

According to the formulas for the momentum and impulse [14,15], the relationship for the instantaneous velocity of the apple during the falling into the fruit box can be calculated as follows:

$$\begin{cases} v_l = \sqrt{2gh_2 + v_t^2} \\ v_l \leq \frac{\int_0^x F(t)dt}{m} \end{cases} \tag{8}$$

where $v_l$ is the instantaneous velocity of the apple upon falling into the fruit box, m/s; $v_t$ is the instantaneous velocity of the apple upon separation from the fruit conveying tube, m/s; $h_2$ is the vertical height of the apple during projectile motion, m; $F(t)$ is the contact stress function during the fruit box landing process, N; $x$ is the unloading time during the fruit box landing process; $m$ is the mass of the apple, kg.

The peel yield force of a Xinjiang Aksu Red Fuji apple harvested when fully mature is $79.91 \pm 2.62$ N. After harvesting, the apples are stored under refrigeration at a temperature of $2 \pm 1$ °C and a relative humidity rate of $90 \pm 5\%$. The apples' hardness shows a gradual declining trend after 39 days of cold storage, and it stabilizes after 45 days of post-ripening cold storage. The peel yield force of apples after 45 days of cold storage is $56.00 \pm 2.58$ N [16]. During the harvesting of fully mature apples, the landing speed into the fruit box typically ranges from 0.4 to 0.8 m/s. Based on Formulas (5), (7) and (8), the selected range for the inclination angle of the fruit conveying tube is $30°$ to $50°$, the selected range for the inner wall radius of the fruit conveying tube is 60 to 100 mm, and the selected range for the length of the fruit conveying tube is 0.12 to 0.22 m.

### 2.5. Energetics Analysis of the Packing Collision Process

The apple packing process can be divided into two stages. The first stage involves the apple rolling along the fruit tray, coming into contact with the inner wall of the fruit conveying tube, and undergoing deformation. When the local stress on the apple reaches its yield limit, the elastic deformation ends and the apple undergoes plastic deformation under the impact force. As the apple's velocity decreases to zero, it reaches the maximum deformation, then the elastic deformation undergoes a rebound, converting some of the elastic potential energy into kinetic energy. This stage is known as the rebound unloading stage. The second stage involves the apple sliding downward along the inner wall of the tube under the influence of elastic potential energy and gravitational potential energy. The apple experiences normal and tangential forces. When the apple separates from the mouth of the fruit conveying tube, all remaining elastic potential energy is converted into kinetic energy and the apple falls into the fruit box, completing the packing operation. This stage is referred to as the sliding friction stage.

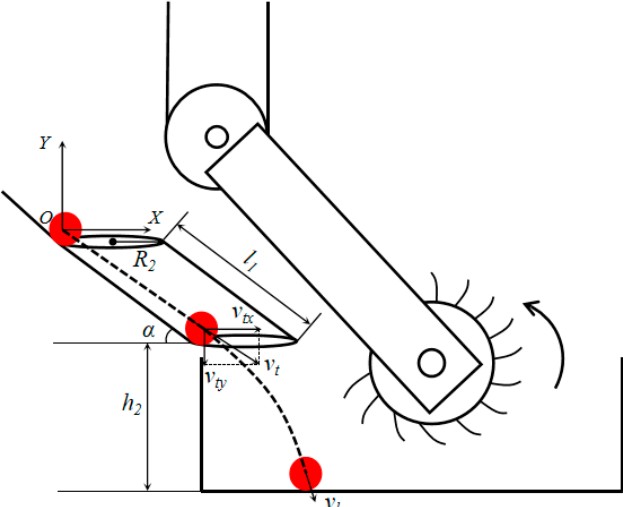

**Figure 4.** A kinematic analysis of an apple sliding along the fruit conveying tube.

The collisions between objects can be classified into two types: direct collision and oblique collision. A direct collision is a special case of an oblique collision, and an oblique collision is a more common scenario in practical situations [17]. In a direct collision, the velocities of the two objects involved are collinear with the line connecting their centers of mass. In an oblique collision, the velocities of the two objects involved are not collinear with the line connecting their centers of mass. When there is a large-angle oblique collision between the apple and the fruit conveying tube, the effect of the tangential force is relatively insignificant compared to the normal force. Therefore, in the first stage of the collision process, the influence of friction on the energy loss can be neglected.

The apple behaves as an elastoplastic material, and the normal plastic deformation in the first stage and the tangential frictional plastic deformation in the second stage are the main contributors to energy loss during the packing process. The plastic deformation of the apple is the primary cause of damage during the packing process.

The normal loading process of the apple is shown in Figure 5. As the apple moves along its velocity direction, it comes into contact with the inner wall of the fruit conveying tube at position 1. The impact force exerted by the apple on the inner wall of the tube gradually increases, causing the apple to undergo elastic deformation. When it reaches position 2, the apple reaches the maximum compression of elastic deformation.

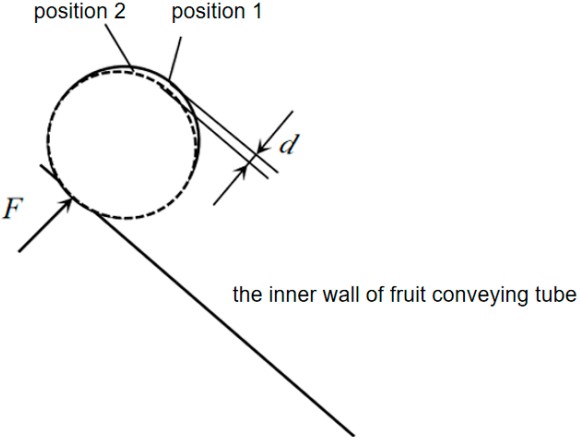

**Figure 5.** The normal loading process.

In order to calculate the yield pressure value of the apple, according to the Hertz contact theory [18,19], the relationship between the normal contact force and the deformation compression is as follows:

$$F = \frac{4}{3} E R^{\frac{1}{2}} d^{\frac{3}{2}} \tag{9}$$

where $F$ is the normal contact force, N; $E$ is the comprehensive elastic modulus, MPa; $R$ is the equivalent radius, mm; $d$ is the deformation compression, mm.

The comprehensive elastic modulus is as follows

$$\frac{1}{E} = \frac{1 - v_1^2}{E_1} + \frac{1 - v_2^2}{E_2} \tag{10}$$

where $E_1$ is the elastic modulus of the apple, MPa; $E_2$ is the elastic modulus of the fruit conveying tube, MPa; $v_1$ is the Poisson's ratio of the apple; $v_2$ is the Poisson's ratio of the fruit conveying tube.

Based on references, the elastic modulus of the apple is determined to be 3.385 MPa with a Poisson's ratio of 0.33. The fruit conveying tube is made of high-density polyethylene material, with an elastic modulus of 680 MPa and a Poisson's ratio of 0.38 [20]. Substituting these values into the formula yields a comprehensive elastic modulus of 3.784 MPa.

The equivalent radius of contact between the apple and the fruit conveying tube is:

$$\frac{1}{R} = \frac{1}{R_1} + \frac{1}{R_2} \tag{11}$$

where $R_1$ is the apple's transverse diameter, mm; $R_2$ is the inner wall radius of the fruit conveying tube in mm.

Xinjiang Aksu Red Fuji apples were selected as the experimental samples. Thirty uniformly sized, intact apples were chosen for the actual measurements. After obtaining the average values, the apples were found to have a transverse diameter of 86 mm and a mass of 0.23 kg. Based on the measured transverse diameter of the apples, the radius of the fruit conveying tube in the low-damage packing device was set to 80 mm. By substituting the data, the equivalent radius was determined to be 28 mm.

The relationship between the yield stress and compressive deformation for elastic plastic materials is as follows:

$$\begin{cases} d = (C\sigma)^2 \left( \frac{\pi \sqrt{R}}{2E} \right)^2 \\ C = min \left( 1.295 e^{0.763 \mu_1}, 1.295 e^{0.763 \mu_2} \right) \end{cases} \tag{12}$$

where $d$ is the yield deformation of the apple, mm; $\sigma$ is the yield stress of the apple, pa; $C$ is the proportionality coefficient, calculated with a value of 1.6.

Substituting Equation (12) into Equation (9), the yield pressure of the apple is calculated as follows:

$$F_1 = \frac{R^2 (\pi C\sigma)^3}{6E^2} \tag{13}$$

The work done by the yield pressure during yielding is:

$$W_1 = \frac{R^3 (\pi C\sigma)^5}{24E^4} \tag{14}$$

The apple slides down from the fruit tray of the low-damage packing device and collides with the inner wall of the fruit conveying tube. In most cases, the collision velocity direction is not collinear with the line connecting the centers of mass of the apple and the tube, indicating an oblique collision [21,22]. Due to the short acceleration time and low velocity of the apple after sliding from the fruit tray, the collision between the apple and the inner wall of the tube is classified as a low-speed collision. The total contact area is

divided into an outer sliding area and an inner adhesive area, with radii areas of '*a*' and '*c*', respectively, as shown in Figure 6.

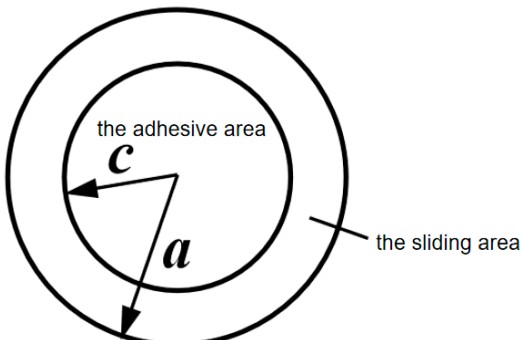

**Figure 6.** The total contact area during the collision.

The expression for tangential force is as follows:

$$F_\tau = \begin{cases} \mu F \left[ 1 - \left( \frac{c}{a} \right)^3 \right], \text{ adhesive} \\ \mu F \qquad\qquad, \text{ sliding} \end{cases} \tag{15}$$

where $F_\tau$ is the tangential force, N; $\mu$ is the coefficient of friction; $a$ is the radius of the outer sliding area, mm; $c$ is the radius of the inner adhesive area, mm.

When two objects come into contact, there is micro-slippage at the adhesive and boundary positions throughout the entire collision process. However, when there is no relative motion between the two objects, the adhesive tangential force formula is used. When there is relative motion and sliding occurs in the contact area, sliding friction force is generated, and in this case the sliding tangential force formula is used. As the apple rolls off the fruit tray and collides with the inner wall of the fruit conveying tube in a slanting manner, the ratio of the initial incident angle to the critical incident angle is greater than 1, indicating relative motion between the two objects. Therefore, the apple undergoes sliding. The initial incident angle values of the apple range from 0 to 90 degrees, and in this study it is taken as 45 degrees for calculations. According to Coulomb's friction law, the tangential force exerted on the apple is calculated as follows:

$$F_\tau = \frac{\sqrt{2} m \mu v_t}{t} \tag{16}$$

The sliding displacement of the contact between the apple and the inner wall of the fruit conveying tube is:

$$l_1 = \sqrt{2} v_t t \tag{17}$$

The work done by the tangential force is as follows:

$$W_2 = 2 e m \mu v_t^2 \tag{18}$$

where $e$ is the coefficient of tangential force restitution, with values ranging from 0 to 1.

During the apple packing process, the low-damage packing device moves upward as the apples accumulate inside the fruit box. The positional height change, denoted as $h_2$, and the instantaneous velocity of the apple as it falls into the fruit box, denoted as $v_l$, remain constant. Therefore, the energy associated with the apple's collision and friction during its entry into the fruit box is limited within a fixed range. Let us define this energy as $W_3$, representing the energy responsible for apple damage during the falling process into the fruit box.

According to the two stages of motion, in which the apple collides with the inner wall of the fruit conveying tube and slides along it, it can be inferred that the kinetic energy

and potential energy of the apple are converted into the work done for elastic deformation, work done by tangential frictional force, and work done for the plastic deformation of the apple. The work done for plastic deformation represents the energy, denoted as $W_4$, responsible for the damage occurring during the apple packing process.

According to the law of conservation of energy [23,24], the energy $W$ responsible for the damage occurring in the apple is calculated as follows:

$$\begin{cases} W = W_3 + W_4 \\ W_4 = mgh_1 + \frac{1}{2}m(v^2 - v_t^2) - \frac{R^3(\pi C\sigma)^5}{24E^4} - 2em\mu v_t^2 \end{cases} \tag{19}$$

Based on the above analysis, it is evident that when factors such as the apple radius, mass, mechanical properties, material of the fruit conveying tube, and mechanical vibrations of the fruit conveying tube during operation are determined, the main influencing factor for the mechanical damage arising from the collision and sliding of the apple along the inner wall of the fruit conveying tube during the packing process is the change in height of the apple's position. To minimize the damage to the apple during the packing process, it is desirable to make $W_4$ approach zero as much as possible, which implies satisfying the following condition:

$$h_1 \leq \frac{1}{2g}\left[\left(v_t^2 - v^2\right) + \frac{R^3(\pi C\sigma)^5}{12mE^4} + 4em\mu v_t^2\right] \tag{20}$$

According to Equation (20), the change in height ($h_1$) during the apple packing process is related to the inclination angle of the fruit conveying tube ($\alpha$), the length ($l_1$) of the fruit conveying tube, and the equivalent contact radius ($R$) between the apple and the fruit conveying tube. In the case of a fixed apple diameter, the equivalent contact radius is only related to the inner wall radius of the tube ($R_2$). Through the literature review, it is known that when the inclination angle of the fruit conveying tube is small, the packing efficiency is low, and the apple is prone to blockage inside the tube. On the other hand, when the inclination angle is large, the apples are more likely to accumulate in the fruit box after sliding down the tube, leading to increased mechanical damage. When determining the range of values for the length of the tube, the distance between the tube outlet and the brush roller should be considered to ensure that the packed apples are dispersed in a timely manner, thereby improving the packing efficiency. Similarly, when determining the range of values for the inner wall radius of the tube, the machine installation position and packing efficiency should be taken into account. If the inner wall radius is too large, this can affect the installation position of the brush roller. Conversely, if the inner wall radius is too small, this can lead to low packing efficiency and potential blockages. Based on the analysis above, the range of values for the inclination angle ($\alpha$) is selected as 30–50°, the range of values for the inner wall radius ($R_2$) is selected as 60–100 mm, and the range of values for the length ($l_1$) is selected as 0.12–0.22 m. Therefore, the following section will utilize an ADAMS kinematic simulation and the orthogonal experimental method to investigate the influence of the inclination angle, the inner wall radius, and the length of the fruit conveying tube on the fruit damage rate and packing efficiency.

## 3. Results

### 3.1. Experimental Materials and Experimental Design

The experimental materials were fifty uniformly sized, intact Xinjiang Aksu Red Fuji apples.

The experimental equipment included a GY-3 fruit hardness tester produced by Quzhou Aipu Metrology Instrument Co., Ltd. (Taian, China), and a CJW vernier caliper produced by Qingdao Airize E-commerce Co., Ltd. (Qingdao, China).

The experimental site was the Shandong Province Horticultural Machinery and Equipment Key Laboratory (Taian, China).

Regarding the experimental method, the samples were measured for their transverse and longitudinal diameters using a vernier caliper, referring to the standard ZB B31007-88 [25,26]. The fruit shape index of the samples was calculated based on Formula (21). Ten samples with fruit shape indices of 0.6 (±0.05), 0.7 (±0.05), and 0.8 (±0.05) were selected. Each sample was carefully peeled with a knife, removing a strip of fruit peel 10 mm wide at the shoulder, trunk, and bottom, minimizing damage to the flesh. The hardness tester was held vertically against each measurement position on the sample, applying pressure until the measuring head reached the specified reference line in the fruit flesh. The reading was recorded from the hardness tester dial. The hardness measurement data for each sample are shown in Table 2. The average value for all samples' hardness was calculated.

**Table 2.** Sample hardness measurement data.

| Serial Number | Apple Hardness/(kg/cm$^2$) | | | | | | | | |
|---|---|---|---|---|---|---|---|---|---|
| | $S = 0.6^{+0.05}_{-0.05}$ | | | $S = 0.7^{+0.05}_{-0.05}$ | | | $S = 0.8^{+0.05}_{-0.05}$ | | |
| | **Shoulder** | **Trunk** | **Bottom** | **Shoulder** | **Trunk** | **Bottom** | **Shoulder** | **Trunk** | **Bottom** |
| 1 | 7.1 | 6.6 | 6.8 | 7.3 | 7.1 | 7.4 | 7.6 | 7.2 | 7.3 |
| 2 | 7.2 | 7.1 | 6.9 | 7.2 | 7.1 | 7.2 | 7.8 | 7.4 | 7.7 |
| 3 | 7.6 | 7.3 | 7.7 | 6.9 | 6.4 | 6.8 | 7.3 | 7.0 | 7.4 |
| 4 | 7.2 | 6.6 | 6.8 | 7.5 | 7.2 | 7.7 | 6.9 | 6.5 | 6.8 |
| 5 | 6.8 | 6.5 | 6.9 | 7.3 | 7.0 | 7.2 | 7.7 | 7.3 | 7.4 |
| 6 | 6.4 | 5.8 | 6.3 | 7.7 | 7.2 | 7.5 | 6.8 | 6.5 | 6.9 |
| 7 | 7.2 | 6.8 | 7.0 | 6.9 | 6.6 | 7.0 | 7.0 | 6.7 | 7.1 |
| 8 | 6.9 | 6.4 | 6.7 | 7.1 | 6.7 | 7.0 | 6.5 | 6.1 | 6.4 |
| 9 | 7.1 | 7.0 | 7.3 | 6.8 | 6.2 | 6.5 | 6.8 | 6.6 | 7.1 |
| 10 | 7.3 | 6.9 | 7.1 | 6.9 | 6.6 | 6.7 | 6.6 | 6.2 | 6.5 |
| variance | 0.17 | | | | | | | | |
| average | 6.97 | | | | | | | | |

The formula for calculating the fruit shape index is as follows:

$$S = \frac{H_1}{H} \tag{21}$$

where $S$ is the fruit shape index; $H_1$ is the fruit's longitudinal diameter, mm; $H$ is the fruit's transverse diameter, mm.

Based on the measurement results obtained from the hardness tester, using the equivalent substitution, where 1 kg/cm$^2$ is equal to 0.1 MPa and the diameter of the pressure head is $\phi 7.9^{+0.02}_{0}$ mm, the yield strength of the apples can be determined. The average yield strength obtained from the measurements was 35.20 N, and this was calibrated as the damage threshold during the apple packing process.

A three-dimensional model for the low-damage packing device for apple harvesting platform was created in SolidWorks 2020 software, and it was imported into ADAMS 2019 simulation software for a kinematic simulation. Firstly, the material properties of the model components were defined, and relevant drives and constraints were added to the model. Then, the material properties and motion parameters of the apple model were defined, and marker points were established at the centroiding point of the apple model. This allowed for the tracking of the marker points' motion trajectory and collision forces during the apple packing process. The conveying simulation experiments are shown in Figure 7.

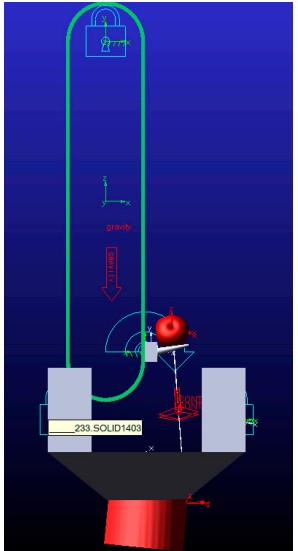 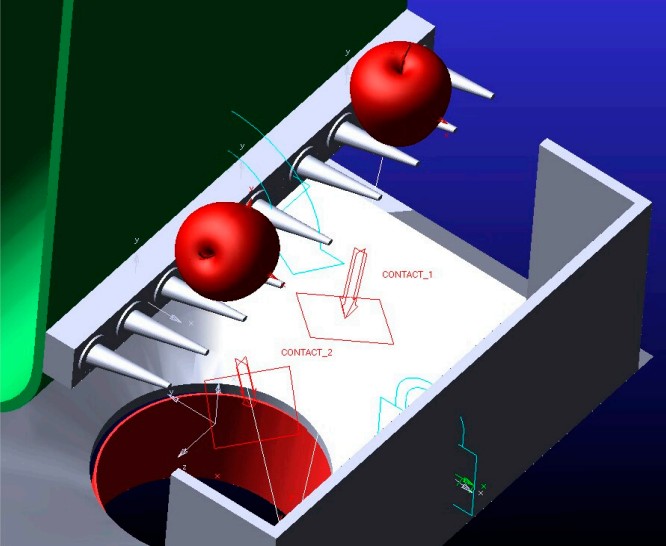

**Figure 7.** The ADAMS simulation experiments.

Multiple repeated simulation experiments were conducted on the apple using the same model parameters during the packing process. By changing the model parameters and repeating the process, the collision forces exerted on the apple and the number of apples packed per unit time under different operating conditions were obtained. The fruit damage rate and packing efficiency during the apple packing process were calculated. The formula for calculating the fruit damage rate is as follows:

$$N = \frac{Z_1}{Z} \times 100\% \tag{22}$$

where $N$ is the fruit damage rate, %; $Z$ is the number of experiments under the same model parameter conditions, 150; $Z_1$ is the number of experiments exceeding the stress damage threshold under the same model parameter conditions.

The formula for calculating the packing efficiency is:

$$I = \frac{m \times n}{t'} \tag{23}$$

where $I$ is the packing efficiency of an apple, kg/h; $m$ represents the average mass of a single apple, taken as 0.23 kg; $n$ represents the number of apples packed under the same model parameters; $t'$ represents the unit time period in hours.

In order to comprehensively evaluate the operational quality for the low-damage packing device of an apple harvesting platform, weight allocation is applied to the two evaluation indicators mentioned above. Due to the different dimensions of these evaluation indicators, it is necessary to perform dimensionless processing on all measured values [27,28]. The formula for dimensionless processing is as follows:

$$X_n^* = \frac{\left| X_n(k) - \overline{X}_n \right|}{\Psi_n} \tag{24}$$

where $X_n(k)$ is the raw data of the $k$-th element for evaluation indicator $n$; $\overline{X}_n$ is the mean value of the same evaluation indicator; $\Psi_n$ is the standard deviation of the same evaluation indicator.

The apple variety and grade are analyzed to determine the weights of the two evaluation indicators, according to the comprehensive evaluation index calculation method for medium-sized Xinjiang Akso Red Fuji apples:

$$Y = 0.55X_N^* + 0.45X_I^* \tag{25}$$

where $X_N^*$ is the dimensionless processing of the fruit damage rate after normalization; $X_I^*$ is the dimensionless processing of the packing efficiency after normalization.

### 3.2. Regression Model Establishment and Validation

To investigate the effects of three factors, namely the inclination angle of the fruit conveying tube (factor $X_1$), the inner wall radius of the fruit conveying tube (factor $X_2$), and the length of the fruit conveying tube (factor $X_3$), on the fruit damage rate and packing efficiency, an orthogonal experimental method was employed for the experiments. The experimental factor coding process is presented in Table 3.

**Table 3.** Experimental factor coding process.

| Encoding Value | Element | | |
| --- | --- | --- | --- |
| | Inclination Angle of the Fruit Conveying Tube $\alpha$/(°) | Inner Wall Radius of the Fruit Conveying Tube $R_2$/(mm) | Length of the Fruit Conveying Tube $l_1$/(m) |
| −1 | 30 | 60 | 0.12 |
| 0 | 40 | 80 | 0.17 |
| 1 | 50 | 100 | 0.22 |

Using Design-Expert 8.0.6 Trial software and based on the Box–Behnken experimental design principle, a response surface experiment with three factors and three levels was designed. The experiment consisted of a total of 17 experimental points, and the specific experimental design plan and response values are presented in Table 4. In the experiment, the values of −1, 0, and 1 for the inclination angle of the fruit conveying tube corresponded to inclination angles of 30°, 40°, and 50°, respectively. Similarly, the values of −1, 0, and 1 for the inner wall radius of the fruit conveying tube corresponded to inner wall radii of 60 mm, 80 mm, and 100 mm, respectively. Lastly, the values of −1, 0, and 1 for the length of the fruit conveying tube corresponded to lengths of 0.12 m, 0.17 m, and 0.22 m, respectively. Regression equations were obtained for the fruit damage rate (*N*) and packing efficiency (*I*) as response values, followed by a significance experiment.

**Table 4.** Experimental design plan and response values.

| Prologue | Inclination Angle of the Fruit Conveying Tube | Inner Wall Radius of the Fruit Conveying Tube | Length of the Fruit Conveying Tube | Fruit Damage Rate (%) | Packing Efficiency (kg/h) |
| --- | --- | --- | --- | --- | --- |
| 1 | 1 | −1 | 0 | 10.7 | 2243 |
| 2 | −1 | 1 | 0 | 8.0 | 2047 |
| 3 | 0 | 1 | −1 | 5.3 | 1571 |
| 4 | −1 | −1 | 0 | 8.3 | 2015 |
| 5 | 0 | 0 | 0 | 6.0 | 1674 |
| 6 | 0 | 0 | 0 | 6.7 | 1722 |
| 7 | 0 | 0 | 0 | 6.3 | 1687 |
| 8 | 0 | 0 | 0 | 6.0 | 1702 |
| 9 | 0 | −1 | 1 | 6.7 | 1831 |

**Table 4.** *Cont.*

| Prologue | Inclination Angle of the Fruit Conveying Tube | Inner Wall Radius of the Fruit Conveying Tube | Length of the Fruit Conveying Tube | Fruit Damage Rate (%) | Packing Efficiency (kg/h) |
|---|---|---|---|---|---|
| 10 | 1 | 1 | 0 | 11.4 | 2444 |
| 11 | 1 | 0 | 1 | 12.7 | 2521 |
| 12 | −1 | 0 | 1 | 8.6 | 2119 |
| 13 | 0 | −1 | −1 | 4.7 | 1492 |
| 14 | −1 | 0 | −1 | 7.6 | 1874 |
| 15 | 0 | 0 | 0 | 6.1 | 1654 |
| 16 | 1 | 0 | −1 | 9.3 | 2197 |
| 17 | 0 | 1 | 1 | 7.3 | 1899 |

Through a multivariate regression analysis using Design-Expert 8.0.6 Trial software, the regression equations were obtained to assess the effects of each factor on the fruit damage rate and packing efficiency after eliminating nonsignificant terms.

$$N = 6.22 + 1.32X_1 + 0.2X_2 + 0.925X_3 + 0.25X_1X_2 + 0.85X_1X_3 + 3.59X_1{}^2 - 0.21X_2{}^2 - 0.01X_3{}^2 \quad (26)$$

$$I = 1687.8 + 168.75X_1 + 47.5X_2 + 154.5X_3 + 42.25X_1X_2 + 19.75X_1X_3 - 2.75X_2X_3 + 489.48X_1{}^2 + 9.97X_2{}^2 + 0.475X_3{}^2 \quad (27)$$

Based on the analysis of Figures 8 and 9, and the $p$-values of each factor and the influences of the factors on fruit damage rate and packing efficiency are as follows, in descending order: the inclination angle of the fruit conveying tube (factor $X_1$), the length of the fruit conveying tube (factor $X_3$), and the inner wall radius of the fruit conveying tube (factor $X_2$). The $p$-value of the fruit damage rate response surface model is less than 0.01, indicating high significance of the regression model. The $p$-value of the lack-of-fit term is greater than 0.1, demonstrating the absence of lack-of-fit factors and indicating a high degree of fit for the regression equation. Therefore, the regression model can be used to replace the actual experimental results for the analysis. The coefficient of determination ($R^2$) is 0.9925, indicating that 99.25% of the variation can be explained by the model, with only 0.75% unexplained variation. This suggests that the model has a good fit and can be used for experimental predictions. Similarly, the $p$-value of the packing efficiency response surface model is less than 0.01, indicating high significance of the regression model. The $p$-value of the lack-of-fit term is greater than 0.1, demonstrating the absence of lack-of-fit factors and indicating a high degree of fit for the regression equation. The coefficient of determination ($R^2$) is 0.9959, indicating that 99.59% of the variation can be explained by the model, with only 0.41% unexplained variation. This further confirms the good fit of the model and its suitability for experimental predictions. An analysis of variance for the regression equations are presented in Table 5.

**Table 5.** An analysis of variance for the regression equations.

| Source of Variance | Fruit Damage Rate (%) | | | | Packing Efficiency (kg/h) | | | |
|---|---|---|---|---|---|---|---|---|
| | Square Sum | Degree of Freedom | F | p | Square Sum | Degree of Freedom | F | p |
| model | 78.75 | 9 | 102.43 | <0.0001 ** | $1.463 \times 10^6$ | 9 | 187.17 | <0.0001 ** |
| $X_1$ | 14.04 | 1 | 164.41 | <0.0001 ** | $2.278 \times 10^5$ | 1 | 262.29 | <0.0001 ** |
| $X_2$ | 0.32 | 1 | 3.75 | 0.0942 | $1.805 \times 10^4$ | 1 | 20.78 | 0.0026 |
| $X_3$ | 6.84 | 1 | 80.13 | <0.0001 ** | $1.910 \times 10^5$ | 1 | 219.86 | <0.0001 ** |
| $X_1X_2$ | 0.25 | 1 | 2.93 | 0.1309 | 7140.25 | 1 | 8.22 | 0.0241 |

**Table 5.** *Cont.*

| Source of Variance | Fruit Damage Rate (%) | | | | Packing Efficiency (kg/h) | | | |
|---|---|---|---|---|---|---|---|---|
| | Square Sum | Degree of Freedom | F | p | Square Sum | Degree of Freedom | F | p |
| $X_1X_3$ | 2.89 | 1 | 33.83 | 0.0007 ** | 1560.25 | 1 | 1.8 | 0.2220 |
| $X_2X_3$ | 0.00 | 1 | 0.00 | 1.0000 | 30.25 | 1 | 0.0348 | 0.8572 |
| $X_1^2$ | 54.27 | 1 | 635.22 | <0.0001 ** | $1.009 \times 10^6$ | 1 | 1161.47 | <0.0001 ** |
| $X_2^2$ | 0.1857 | 1 | 2.17 | 0.1839 | 418.95 | 1 | 0.4824 | 0.5097 |
| $X_3^2$ | 0.0004 | 1 | 0.0049 | 0.946 | 0.95 | 1 | 0.0011 | 0.9745 |
| Residue | 0.598 | 7 | | | 6079.8 | 7 | | |
| Loss of fit value | 0.25 | 3 | 0.9579 | 0.4937 | 3375 | 3 | 1.66 | 0.3104 |
| Error | 0.348 | 4 | | | 2704.8 | 4 | | |
| Synthesize | 79.35 | 16 | | | $1.469 \times 10^6$ | 16 | | |

Notes: ** indicates a significant difference ($p < 0.01$).

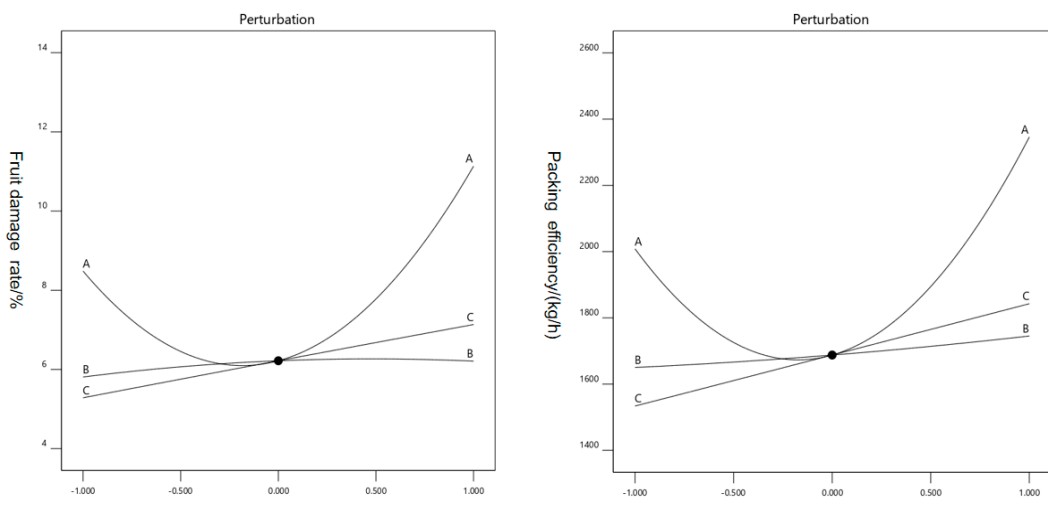

**Figure 8.** Deviation of coded factors from the reference point for the fruit damage rate and packing efficiency. Notes: A is the inclination angle of the fruit conveying tube; B is the inner wall radius of the fruit conveying tube; C is the inclination angle of the fruit conveying tube.

### 3.3. Analysis of Model Interaction Terms

Based on the regression model, response surface graphs were created to depict the relationships between the fruit damage rate, packing efficiency, and various factors. The shape of the response surface reflects the strength of the interaction between the factors.

Figure 10a demonstrates the interactive effects of the inclination angle and the length of the fruit conveying tube on the fruit damage rate. When the length remains constant, the fruit damage rate exhibits a trend of initially decreasing and then increasing with an increase in the inclination angle. Additionally, when the length is long, the influence of the inclination angle on the fruit damage rate is more significant, as indicated by the steep curve in the graph. This suggests that appropriately reducing the inclination angle can significantly decrease the fruit damage rate when the length varies within the range of 0.17–0.22 m. The main reason for this is that under the condition of a constant length, the frictional force exerted on the apples by the fruit conveying tube is determined by the inclination angle. During the sliding process of the apples along the fruit conveying tube, the frictional energy loss and damage are reduced, resulting in a lower fruit damage rate. The shape of the response surface and the density of the contour lines indicate that the effect of the inclination angle on the fruit damage rate is greater than that of the length, which aligns with the results of the variance analysis.

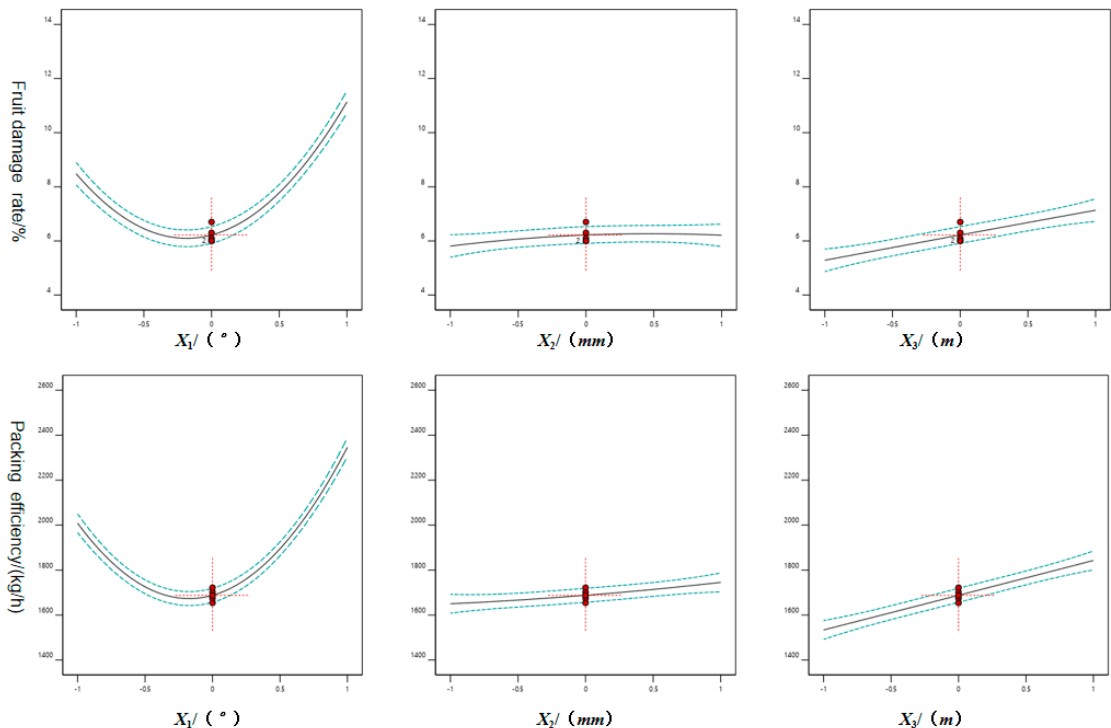

**Figure 9.** Deviation of single factors from the reference point for the fruit damage rate and packing efficiency.

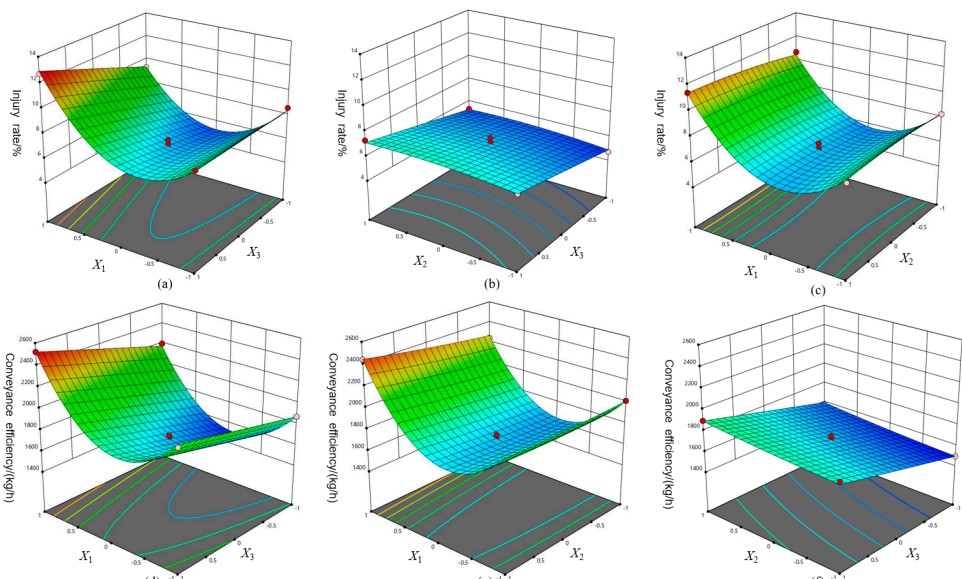

**Figure 10.** (**a–f**) Response surfaces of the interactive effects of factors on the fruit damage rate and packing efficiency.

Figure 10b illustrates the interactive effects of the inner wall radius and length of the fruit conveying tube on the fruit damage rate. When the length remains constant, the fruit damage rate slightly increases with an increase in the inner wall radius. Conversely, when the inner wall radius is fixed, the fruit damage rate decreases with a reduction in length. Furthermore, when the inner wall radius is at a high level, the influence of the length on the fruit damage rate is more significant, as depicted by the steep curve in the graph. This indicates that when the inner wall radius values vary within the range of 80–100 mm, appropriately reducing the length can significantly lower the fruit damage

rate. The primary reason for this is that under the condition of a constant inner wall radius, the frictional force exerted on the apples by the tube is determined by the length. During the sliding process of the apples along the tube, the frictional energy loss and damage are reduced, resulting in a lower fruit damage rate. The shape of the response surface and the density of the contour lines indicate that the two factors have different weights in terms of their impact on the fruit damage rate. The length has a greater influence on the fruit damage rate compared to the inner wall radius, which aligns with the results of the variance analysis.

Figure 10c displays the interactive effects of the inclination angle and the inner wall radius of the fruit conveying tube on the fruit damage rate. When the inner wall radius is constant, the fruit damage rate shows a decreasing trend followed by an increasing trend with an increase in the inclination angle. On the other hand, when the inclination angle is constant, the fruit damage rate slightly increases with an increase in the inner wall radius. The main reason for this can be attributed to the fact that when the inclination angle is fixed at a certain level, the apples acquire a lower initial velocity and kinetic energy value, resulting in lower collision stress and a lower fruit damage rate when falling into the fruit box under the effect of gravity. Additionally, when the inner wall radius is at a moderate level, the apple packing process is less prone to blockages and exhibits a well-designed mechanical structure. The shape of the response surface and the contour density indicate that the interaction between the inclination angle and inner wall radius of the fruit conveying tube have a significant impact on the fruit damage rate. Furthermore, the inclination angle has a greater influence on the fruit damage rate than the inner wall radius, which is consistent with the results of the variance analysis.

Figure 10d demonstrates the interactive effects of the inclination angle and the length of the fruit conveying tube on the packing efficiency. When the length remains constant, the packing efficiency exhibits a trend of initially decreasing and then increasing with an increase in the inclination angle. Moreover, when the length is long, the influence of the inclination angle on the packing efficiency is more significant, as indicated by the steep curve in the graph. This indicates that within the range of moderate to long lengths, appropriately increasing the inclination angle can significantly improve the packing efficiency. The primary reason for this is that when the inclination angle is at a high level, the buffering resistance exerted on the apples during the rolling process into the container is reduced, allowing the apples to roll more quickly into the container, thereby improving the packing efficiency. The shape of the response surface and the density of the contour lines indicate that the interactive effect of the inclination angle and the length has a significant impact on the packing efficiency. The inclination angle has a greater influence on the packing efficiency compared to the length, which aligns with the results of the variance analysis.

Figure 10e demonstrates the interactive effects of the inclination angle and inner wall radius of the fruit conveying tube on the packing efficiency. When the inner wall radius remains constant, the packing efficiency exhibits a trend of initially decreasing and then increasing with an increase in the inclination angle. Conversely, when the inclination angle is fixed, the packing efficiency slightly increases with an increase in the inner wall radius. The primary reason for this is that with the inclination angle fixed at a certain level, a decrease in the inner wall radius increases the resistance experienced by the apples as they pass through the fruit conveying tube, leading to a higher likelihood of apple blockage and lower packing efficiency. The shape of the response surface and the density of the contour lines indicate that the inclination angle has a greater impact on the packing efficiency, which aligns with the results of the variance analysis.

Figure 10f demonstrates the interactive effects of the inner wall radius and length of the fruit conveying tube on the packing efficiency. When the length remains constant, the packing efficiency slightly increases with an increase in the inner wall radius. Conversely, when the inner wall radius is fixed, the packing efficiency increases with an increase in the length, with the latter showing a more significant change. The two factors have

different weights in terms of their impact on the packing efficiency, with the length having a greater influence compared to the inner wall radius, which aligns with the results of the variance analysis.

### 3.4. Experimental Parameter Optimization

To optimize and improve the fruit damage rate and packing efficiency of apples in the apple harvesting platform's packing operation process, the optimization function of the Design-Expert 8.0.6 Trial software was applied. The regression equation models for the influencing factors and performance indicators were solved through optimization. Based on the actual working conditions of the low-damage packing device for the apple harvesting platform, the operational performance requirements, and the aforementioned regression models, an analysis was conducted. The optimization objective functions were selected as the fruit damage rate and packing efficiency, with the inclination angle, the inner wall radius, and the length of the fruit conveying tube acting as the constraint conditions. The mathematical model is as follows:

$$\begin{cases} minN(x_1, x_2, x_3) \\ maxI(x_1, x_2, x_3) \\ s.t. \begin{cases} 37.3° \le x_1 \le 47.2° \\ 72.5 \text{ mm} \le x_2 \le 87.5 \text{ mm} \\ 0.12 \text{ m} \le x_3 \le 0.15 \text{ m} \end{cases} \end{cases} \tag{28}$$

By performing optimization calculations, the optimal parameter combination was obtained; when the inclination angle of the fruit conveying tube was 47.1°, the inner wall radius was 83.9 mm and the length was 0.12 m, and the low-damage packing device achieved the best combination of the fruit damage rate and packing efficiency for apple packing. The resulting fruit damage rate was 7.5% and the packing efficiency was 1902 kg/h.

### 3.5. Verification Experiment

The experimental materials were fifty uniformly sized, intact Xinjiang Aksu Red Fuji apples.

The experimental equipment was the low-damage packing device processed by Taian City Shanli Machinery Equipment Co., Ltd. (Taian, China).

The experimental site was the Shandong Province Horticultural Machinery and Equipment Key Laboratory (Taian, China).

Regarding the experimental method, the validation experiment method was consistent with the orthogonal experiment. Following the evaluation method of ZB B31007-88, a validation experiment was conducted in March 2023 at the Key Laboratory of Horticultural Machinery and Equipment at Shandong Agricultural University. The line speed for the vertical conveyance device was set to 0.2 m/s. Based on the optimization results from the model, the experimental parameters were adjusted accordingly. The optimized operating parameters for the device are an inclination angle of 47° for the fruit conveying tube, an inner wall radius of 84 mm, and a length of 0.12 m for the fruit conveying tube. Under the same parameter conditions, five repeated experiments were conducted to determine the fruit damage rate and packing efficiency after parameter optimization, thereby validating the performance of the low-damage packing device. The packing experiment for the apple harvesting platform's low-damage packing device after parameter optimization is illustrated in Figure 11, with the results detailed in Table 6.

According to Table 6, when the inclination angle of the fruit conveying tube was 47°, the inner wall radius of the fruit conveying tube was 84 mm, the length of the fruit conveying tube was 0.12 m, the average fruit damage rate of the low-damage packing device was 7.2%, and the average packing efficiency was 1925 kg/h. The relative error rates between the experimental values and the optimized values for the fruit damage rate and packing efficiency were 0.3% and 1.2%, respectively, both of which were less than 5%. This indicates that the regression model is accurate. The low-damage packing device meets

the requirements for orchard packing operations, and the research findings can provide references for optimizing the packing operation parameters of apple harvesting platforms.

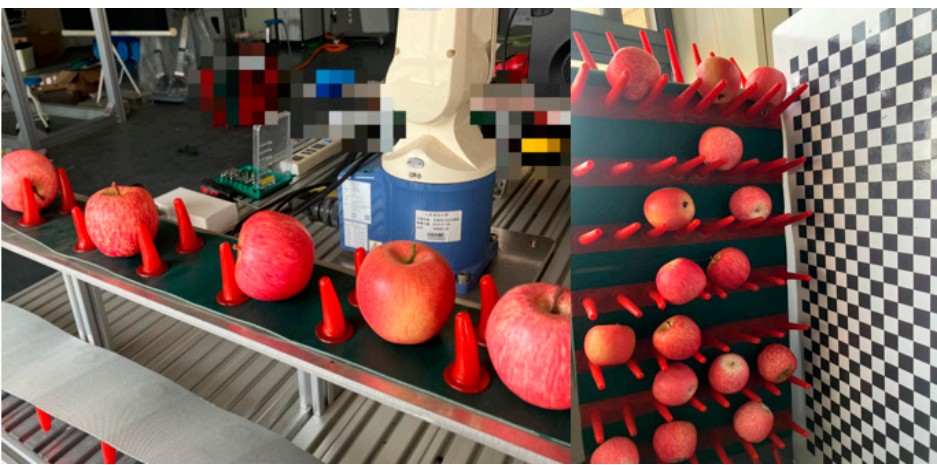

**Figure 11.** Experimental diagram of the low-damage packing device for apple packing.

**Table 6.** Experimental results for the low-damage packing device used for apple packing.

| Number of Experimental Trials | Inclination Angle of the Fruit Conveying Tube $\alpha/°$ | Inner Wall Radius of the Fruit Conveying Tube $R_2$/mm | Length of the Fruit Conveying Tube $l_1$/m | Fruit Damage Rate N/% | Packing Efficiency I/(kg/h) |
|---|---|---|---|---|---|
| 1 | 47 | 84 | 0.12 | 7.22 | 1948.8 |
| 2 | 47 | 84 | 0.12 | 7.12 | 1884.8 |
| 3 | 47 | 84 | 0.12 | 7.08 | 1888.5 |
| 4 | 47 | 84 | 0.12 | 7.45 | 1993.0 |
| 5 | 47 | 84 | 0.12 | 7.24 | 1911.6 |

## 4. Discussion

The apple harvesting platform with the low-damage packing device designed and optimized in this study enables high-quality apple harvesting. It is suitable for apple orchards adopting the dwarfing rootstock and high-density planting mode commonly used in China. Compared to large apple harvesting platforms used in the United States and other countries, this harvesting platform is compact in size and does not require a high-power unit. In comparison to other domestic harvesting platforms, it has the advantages of a simple structure, low fruit damage rate, and high packing efficiency. In future work, further optimization of the structure and working parameters of the low-damage packing device on the harvesting platform should be carried out to enhance the packing efficiency and reduce the fruit damage rate.

## 5. Conclusions

By conducting kinematic and energy analyses of the packing process for the low-damage packing device of an apple harvesting platform, a mathematical model for apple energy damage was established. It was determined that the main factors influencing the fruit damage rate and packing efficiency during the apple packing process are the inclination angle, the inner wall radius, and the length of the fruit conveying tube.

A simulation experiment bench model for the low-damage packing device for apples was designed, with the inclination angle, the inner wall radius, and the length of the fruit conveying tube acting as independent variables, and the fruit damage rate and packing efficiency acting as response values. A regression mathematical model was established to describe the relationship between the independent variables and the response values. The

analysis revealed that the factors with the greatest impact, in descending order, were the inclination angle, the length, and the inner wall radius of the fruit conveying tube.

Based on the regression mathematical model of the independent variables and response values obtained from the parameter optimization, the apple packing validation experiment was conducted. The experimental results showed that when the inclination angle of the fruit conveying tube was 47°, the inner wall radius of the fruit conveying tube was 84 mm, the length of the fruit conveying tube was 0.12 m, the average fruit damage rate was minimized at 7.2%, and the average packing efficiency was maximized at 1925 kg/h.

**Author Contributions:** Writing—original draft preparation, Z.C., H.Z. and J.W.; writing—review and editing, Z.C. and H.Z.; conceptualization, Z.C., H.Z. and J.W.; methodology, Z.C., H.Y., Y.Y. and J.S.; software, Z.C., H.Z., G.Z. and G.F.; validation, Z.C., H.Z., H.Y. and J.W.; resources, J.W.; data curation, Z.C. All authors have read and agreed to the published version of the manuscript.

**Funding:** This research were funded by China Agriculture Research System (CARS-27) and Shandong Province Key Research and Development Plan (2022CXGC020701).

**Institutional Review Board Statement:** Not applicable.

**Informed Consent Statement:** Not applicable.

**Data Availability Statement:** The data are available within the article.

**Acknowledgments:** Thanks to all the authors cited in this article and the referee for their helpful comments and suggestions.

**Conflicts of Interest:** The authors declare no conflict of interest.

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
