# Peer review of "Optimization and Experimental Study of Structural Parameters for a Low-Damage Packing Device on an Apple Harvesting Platform"

_agriculture, doi:10.3390/agriculture13091653_

Round 1

Reviewer 1 Report

This article presents a kinematic and energetic analysis of the apple packing process and develops a damage energy mathematical model to determine the main factors affecting mechanical damage in apples and their range. Parameter optimal experiments were carried out using Adams software to determine the low-damage packaging and high-efficiency device of the apple harvesting platform. The optimal inclination angle of the fruit conveyor tube, the inner wall radius of the fruit conveyor tube and the length of the fruit conveyor tube were obtained. The tables and overall formatting of the article is excellent and the language flows well.

However, there are still some details to be improved:

1. L91: “belt, a fruit conveying tube, fruit trays and brush rollers, as shown in Figure 1”.

       Figure 1 is not clear enough, as the location of "fruit trays" is not found in the figure.

2. L123: “The process of apple damage can be summarized as the collision between the apple and the upper end of the fruit conveying tube, and the friction between the apple and the inner wall of the fruit conveying tube during sliding” .

       The collision damage of an apple falling out of a fruit box should be considered as well.

3. L138: “The line speed range for the low-damage packing device is 0-0.6 m/s”.    The position of "line speed" is not found in Figure 3.

4. L149: “??=√2?0?????−(?????)2”.

No specific explanation was found for v in the formula.

5. L369: “Based on the analysis and preliminary experimental results…”.

More references to analyses above, no preliminary experimental results in detail.

6. L427: “ mass of a single apple, taken as 0.23 kg; n—the represents the number of apples packed” is inconsistent with the mass of apples in L280”the apples were found to have a transverse diameter of 86mm and a mass of 279g.”

7. The size of apples is too homogeneous, and it has not been demonstrated whether different sizes of apples in the orchard can achieve minimum damage and maximum efficiency. Only for Fuji apples in Xinjiang, but not for other kinds of apples, whether it has the generality needs to be studied.

Improve the English language, try to use simple sentences that describe the research process objectively. Check the variables and formulas in the full text, whether the variables have been explained one by one.

Reviewer 2 Report

The problem studied in this paper is relevant and interesting. Overall, the manuscript is well organized and its presentation is good. However, some major issues still need to be improved:

(1) The current literature review is insufficient. The author needs to improve in the introduction, and clarify the innovations of this article. Besides, the citation format of many references is incorrect, such as [1,2,3,4], [5,6,7,8]. Regarding the literature survey, the reviewer recommends to add a few more papers related to this study. Such as: En Lu, Zhongming Tian, Lizhang Xu, Zheng Ma, Chengming Luo. Observer-based robust cooperative formation tracking control for multiple combine harvesters [J]. Nonlinear Dynamics, 2023, 111: 15109-15125. https://doi.org/10.1007/s11071-023-08661-x.

(2) In the analysis of the apple packing collision process in Section 3, the author should provide details of parameter selection for readers to easily repeat. Besides, the author should consider more about the motion state of the apple and the force changes at different stages.

(3) What is the process of parameter optimization in section 4.4?

(4) The experimental verification in section 4.5 lacks a description of experimental conditions and testing techniques, and the author should supplement relevant content.

(5) The content of the paper needs to be slightly streamlined and more focused on the core content. The English needs to be improved sufficiently. Besides, there are many formatting errors in the paper.

The English needs to be improved sufficiently.

Round 2

Reviewer 2 Report

This paper can be accepted.

Minor editing of English language required.